

# Seasonal variability of atmospheric tides in the mesosphere and lower thermosphere: meteor radar data and simulations

Dimitry Pokhotelov[1], Erich Becker[1], Gunter Stober[1], and Jorge L. Chau[1]

[1]Leibniz-Institute of Atmospheric Physics at the University of Rostock, Kühlungsborn, Germany

**Correspondence:** D. Pokhotelov (pokhotelov@iap-kborn.de)

**Abstract.** Thermal tides play an important role in the global atmospheric dynamics and provide a key mechanism for the forcing of thermosphere/ionosphere dynamics from below. A novel method for extracting tidal contributions, based on the adaptive filtering, is applied to analyse multi-year observations of mesospheric winds from ground-based meteor radars located in Northern Germany and Norway. The observed seasonal variability of tides is compared to simulations with the Kühlungsborn

Mechanistic Circulation Model (KMCM). It is demonstrated that the model provides reasonable representation of the tidal amplitudes. The limitations of applying a conventionally coarse resolution model in combination with parametrisation of gravity waves are discussed. The work is aimed towards the development of an ionospheric model driven by the dynamics of the KMCM.

## 1    Introduction

The region of mesosphere and lower thermosphere (MLT) is characterised by a variety of waves including atmospheric gravity waves (GWs), tides, and planetary waves (PWs). In the MLT region these waves reach large amplitudes such that the velocity perturbations become comparable to velocities of the mean flow. While GWs generally break in the MLT region, the tides propagate directly to higher altitudes and impact the dynamics of the thermosphere and ionosphere. The tides thus play an important role in the forcing of the coupled ionosphere-thermosphere system from below (e.g., Yiğit and Medvedev, 2015). Pronounced

features of the low-latitude ionospheric dynamics, such as the wave-4 longitudinal structure observed in sub-equatorial ionospheric electric fields and plasma densities, have been attributed to the forcing from atmospheric tides (Immel, 2006; England et al., 2010). The current work is motivated by the need to simulate the tidal dynamics in the MLT with a computationally inexpensive general circulation circulation model (GCM), and to drive an ionospheric model with the simulated dynamical fields in order to analyse the impact of tides on the ionosphere. Multi-year observations of tides with ground-based meteor

radars are used here as a benchmark for the GCM results.

The thermal tides observed in the MLT region represent an interference of the sun-synchronous (migrating) tides generated by the absorption of infra-red and ultra-violet solar radiation in the troposphere and stratosphere, and the non-sun-synchronous (non-migrating) tides generated by the longitudinal irregularities in radiative heating and latent heat release in troposphere and/or by nonlinear interactions between PWs and migrating tides (e.g., Hagan and Forbes, 2002). The most prominent spectral

components are 24-hour (diurnal), 12-hour (semidiurnal) and 8-hour (terdiurnal) tides. A number of observational studies using



ground-based very high frequency (VHF) meteor radars have been dedicated to the seasonal variability of atmospheric tides in the MLT region[1]. At low latitudes, the diurnal tide dominates the spectrum. It's annual cycle shows minimum amplitudes around the solstices and maximum amplitudes around the equinoxes (e.g., Buriti, 2008; Davis et al., 2013). At middle and high latitudes, the diurnal tides cannot effectively propagate into the MLT region (Lindzen and Chapman, 1969), and the spectrum is dominated by the semidiurnal tide, with the highest amplitudes in winter months and during the fall transition in September (e.g., Mitchell et al., 2002; Manson et al., 2009; Hoffmann et al., 2010; Jacobi, 2012).

Comprehensive whole atmosphere GCMs such as the Canadian Middle Atmosphere Model (CMAM), the Hamburg Model of the Neutral and Ionized Atmosphere (HAMMONIA), or the Whole Atmosphere Community Climate Model (WACCM) reproduce, to some extent, the climatology of the diurnal tide as observed by satellites (McLandress, 2002; Achatz et al., 2008; Smith, 2012). A substantial work on the modelling of tides was also done using the Global Scale Wave Model (GSWM) (Hagan and Forbes, 2002). In GSWM, however, the nonlinear tidal dynamics and interactions with PWs and GWs are neglected. Comparisons of model results focused mainly on satellite observations yielding tidal amplitudes that are averaged over typically 2 months (Forbes et al., 2006; Oberheide et al., 2006). (Mitchell et al., 2002) and (Davis et al., 2013) presented comparisons with model simulations (using the GSWM, CMAM, and WACCM) and meteor radar observations of tides at high and low latitudes, respectively. In these studies the comparison was done including the observed monthly variabilities of tidal amplitudes making the model comparison somewhat inconclusive, as the observed monthly variabilities are comparable to the absolute values of tidal amplitudes.

In the current article we present a comparison between the tidal amplitudes observed with meteor radars at middle and high latitudes, extracted using the novel adaptive filtering algorithm, and the simulated tidal amplitudes extracted from the Kühlungsborn Mechanistic Circulation Model (KMCM) using the same filtering algorithm. This allows a direct comparison of the observed tidal amplitudes with the modelled results, without results being contaminated by the monthly variabilities of the tides.

## 2 Radar observations and data analysis

VHF meteor radars provide neutral wind dynamics in the range of altitude between about 75 and 110 km using backscatter from meteor ionisation traces. The IAP Radar Remote Sensing Department have been continuously operating meteor radars for over a decade at high- and mid-latitude locations in Andenes, Norway (69°N 16°E) and in Juliusruh, Germany (54°N 13°E). In this study the composite tidal climatologies are derived from the datasets of years 2003–2016 for Andenes and November 2007–2016 for Juliusruh.

In order to separate contributions from diurnal, semidiurnal and terdiurnal tidal components, the 1-hour time resolution meteor radar data are processed using an adaptive spectral filter, which uses a sliding window of a predefined length and fits the amplitudes and phases for each tidal component accounting for the number of wave cycles within the window (Stober et al., 2017). The fitting procedure also eliminates the contribution of PWs. The GW contribution is then defined by the residuum

---

[1]A review of the studies with medium frequency radars and other ground instruments covering lower ranges of altitudes is beyond the scope of this article.





and contains all fluctuations different from the tides or PWs. Figures 1 and 2 present tidal climatologies of semidiurnal tides for Andenes and Juliusruh, respectively.

## 3 Numerical simulations

The KMCM is a mechanistic GCM from the surface to the lower thermosphere with uppermost level around $8 \times 10^{-7}$ hPa,
corresponding to about 200 km height. Here we use the same model version as in Becker (2017). This model simulates the dynamics of the whole atmosphere like a comprehensive GCM. The mechanistic character is due to simplified computations of radiative transfer and moist convection, as well as due to the neglect of chemical processes in the middle atmosphere. This mechanistic approach allows the easy adjustment of model parametrisations in order to perform sensitivity experiments. The only ionospheric process considered is a simple parametrisation of ion drag (Becker, 2017). Since the model employs a
conventionally coarse spatial resolution (spectral truncation at a total horizontal wave number 32 and 80 vertical layers), both orographic and non-orographic GWs need to be parametrised.

At the locations corresponding to Andenes and Juliusruh, the model time series are extracted and converted from pressure levels to geometric heights. The same tidal amplitude analysis as for the meteor radar data is applied to the model data. The resulting semidiurnal tidal amplitudes of the zonal and meridional winds simulated with the model are shown in Figures 3 and
4, for Andenes and Juliusruh, respectively. In the following we compare these results with the tidal climatology from the radar winds.

## 4 Discussion and summary

As expected from the linear tidal theory (Lindzen and Chapman, 1969), as well as from earlier observational and modelling studies, the MLT tidal spectra at middle and high latitudes are dominated by the semidiurnal tide (the diurnal and terdiurnal tide
are much weaker, not shown here). The annual cycle of the semidiurnal tide, both at Andenes and Juliusruh, shows maximum amplitudes in winter months (December-February) and during the fall transition in September, while minimum amplitudes are seen ~1 month prior to the summer and winter solstices, i.e. in May and in November. The tidal amplitudes are $\sim 30\%$ stronger at middle latitudes (Juliusruh) than at high latitudes (Andenes).

The simulated tides show similar behaviour as the radar-observed tides. In particular, the highest amplitudes occur in winter
and during the fall transition. Stronger tidal amplitudes at middle than at high latitudes, as well as stronger tidal amplitudes of the meridional than the zonal component, are also reproduced in the simulation. The main difference between the observed and simulated behaviour is that the model predicts strong amplitudes around 80–85 km in the summer months, which is not seen in the observations.

The tides are strongly affected by the interactions with mean winds and GWs (McLandress, 2002; Becker, 2017). A com-
parison between the observed and simulated mean zonal winds, obtained by 21-day time averaging (see Figure 5) shows that the mesopause wind reversal reproduced in the model is too low in altitude by $\sim 5$ km and that the eastward winds higher



up are strongly overestimated comparing to the observational result. This model deficiency might contribute to the simulated amplification of the tides in summer. Moreover, the effects of GWs on both the mean flow and on the amplitudes of tides could play an important role, but the details are difficult to assess. While the GW climatologies can be derived from the radar observations (see Section 2), the same analysis cannot be directly applied to the model where GWs are parametrised. A conventional

coarse-resolution GCM (like the current KMCM) will always produce some resolved inertia-GW activity at MLT altitudes (e.g., Shepherd et al., 2000; McLandress et al., 2006), and these GWs are strongly resolution-dependent. An approximately realistic representation of GWs in a GCM would require effective horizontal and vertical resolutions of less than $\sim 100$ km and 1 km, respectively. A new version of the KMCM allows to perform such simulations with realistic GW effects in the middle atmosphere that are solely due to resolved GWs (Becker and Vadas, 2018). However, a comparison of these model results with

observations is beyond the scope of the present study.

We have demonstrated that the KMCM used with a conventional model setup provides a reasonable representation of the annual cycle of the semidiurnal tide in MLT region at middle and high latitudes. This opens a pathway for the simulation of tidal influence on the thermosphere and ionosphere by coupling the KMCM dynamics to a dedicated model of ionospheric dynamics, specifically the Thermosphere-Ionosphere-Electrodynamics General Circulation Model (TIEGCM) (Maute, 2017).

In this setup the ionospheric model would be forced at its lower boundary located at $\sim$97 km altitude by the GCM dynamical fields at that altitude. Therefore, the presented validation of model dynamics, and tides in particular, with meteor radars in this altitude range is of particular interest. In this respect, the current work represents a first step towards the analysis of tidal forcing of the ionosphere from below.

**Acknowledgements**

This work is partially supported by the Deutsche Forschungsgemeinschaft (DFG, German Research Foundation) under the SPP 1788 (DynamicEarth) Project DYNAMITE (CH 1482/1-1) and by the WATILA Project (SAW-2015-IAP-1 383). We thank the colleagues of the tidal matrix group at IAP for helpful discussions.





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

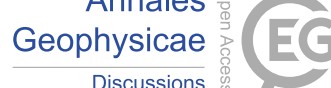



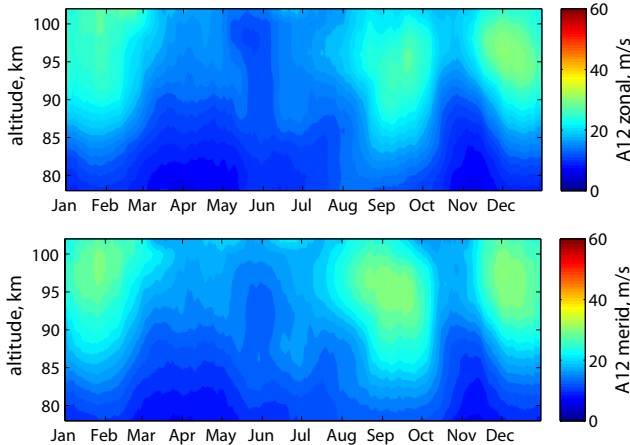

**Figure 1.** Amplitudes of semidiurnal tides at high latitudes extracted from meteor radar observations over Andenes. The top and bottom panels correspond, respectively, to the zonal and meridional components.

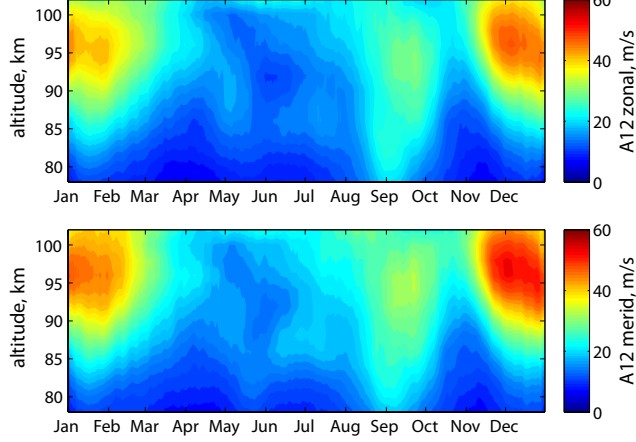

**Figure 2.** Amplitudes of semidiurnal tides at middle latitudes extracted from meteor radar observations over Juliusruh. The top and bottom panels correspond, respectively, to the zonal and meridional components.




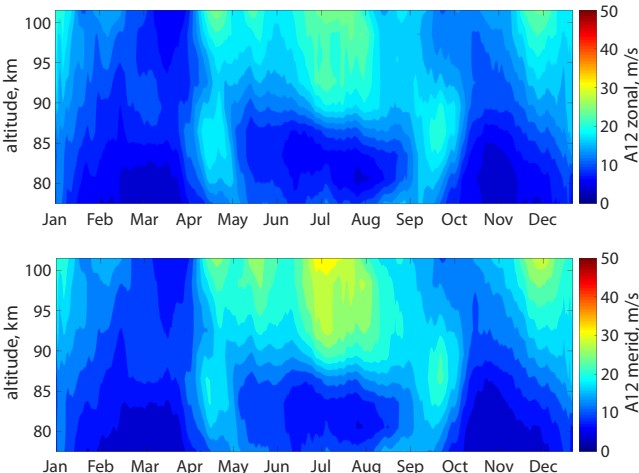

**Figure 3.** Amplitudes of semidiurnal tides at high latitudes (corresponding to Andenes) extracted from the KMCM simulation. The top and bottom panels correspond, respectively, to the zonal and meridional components.

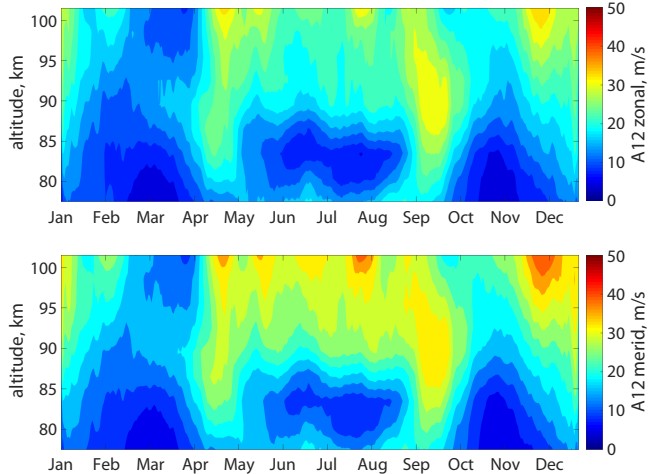

**Figure 4.** Amplitudes of semidiurnal tides at middle latitudes (corresponding to Juliusruh) extracted from the KMCM simulation. The top and bottom panels correspond, respectively, to the zonal and meridional components.





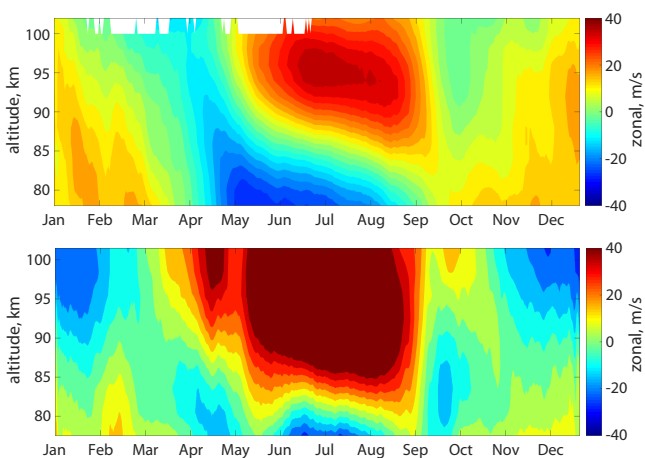

**Figure 5.** Zonal component of the mean flow observed with meteor radar at Juliusruh (top panel) and simulated with the KMCM over the same location (bottom panel). White bins in the top panel reflect insufficient statistics of the observed meteor echoes at high altitudes.