# Peer review of "Seasonal variability of atmospheric tides in the mesosphere and lower thermosphere: meteor radar data and simulations"

_Annales Geophysicae, 2018_

## Referee Comment (RC1) · Anonymous Referee #1 · 9 Mar 2018

Review on paper

Seasonal variability of atmospheric tides in the mesosphere and lower thermosphere and lower thermosphere: meteor radar data and simulations

by

Dimitry Pokhotelov, Erich Becker, Gunter Stober and Jorge L. Chau

This paper deals with a comparison of the measured amplitudes of the semidiurnal tides in the horizontal winds (zonal and meridional) obtained as an average annual behavior from 75- to 110 km altitude at two stations with the simulated results of the

Kühlungsborn Mechanistic Circulation Model (KMCM). Two long wind data series obtained by Meteor Radar at a mid-latitude site Juliusruh, Germany (54o N, 13oE) for nearly 10 years and at a high-latitude site Andenes, Norway (69oN, 16oE) for nearly 14 years are used to extract the tidal components. It is said that a novel adaptive filter method is used for extracting the tidal components from the data series and that the same treatment is applied to the model run results. The authors claim to demonstrate that the model provide reasonable representation of the tidal amplitude and concludes that the coarse spatial resolution model version of KMCM can be used to dynamically force an ionospheric model. The paper is concise, well written and logically consistent and can have interest to Annales Geophysicae readers. However, this reviewer has some concerns about the comparisons presented and their interpretation. Therefore, major revision is needed before its publication. The main concern refers to the way in with the similarities are stressed (referring to the comparison of figures 1 and 2 with figures 3 and 4) and the differences are minimized. Although the main characteristic of the seasonal variation of the amplitudes of semidiurnal tides are captured by the model it is only said that "The main difference between the observed and simulated behaviour is that the model predicts strong amplitudes around 80-85 km in summer months, which is not seen in the observations". A simple analysis of the figures show that the model simulation shows an enhancement on April-May in all altitudes and enhanced amplitudes above 85 km from May to September in mid-and high-latitudes which are not observed on data. There is no discussion how this drawback could affect the use of this conventional model setup to analyze tidal impact in the ionosphere. Technical corrections: Abstract and Page 2, line 19: A "novel" method. This word causes an impact, but as shown in the paper, the method has already been described in a previous work. Page 2, line 2. The footnote 1 is dispensable. Page 2, line 25: The whole meaning of IAP should be described. Page 2, line 25: ..has been continuously... Page 2, lines 31,32. Better put in text that the used method is given by Stober et al., 2017. Page 3, line 27: .. strong amplitudes around 80-85 km in summer months... We can not see this. See special comment above.

---

## Referee Comment (RC2) · Anonymous Referee #2 · 20 Mar 2018

Review of "Seasonal variability of atmospheric tides in the mesophere an lower ther-mosphere: meteor radar data and simulations" by Dimitry Pokhotelov et al. (ANGEO communicates, March 2018)

This paper compares the semi-diurnal tide amplitudes from two meteor radars at north-ern latitudes with the Kühlungsborn Mechanistic Climate Model (KMCM). It is well writ-ten and well organized.

**1.** This is a very light paper as regards content: what is the purpose? If the meteor data are taken as representing the true semi-diurnal tide, then it seems the purpose is to show the quality of the mesospheric KMCM. In this case, comparison of the MWR with other tidal models, e.g. GSWM, is essential. That is, are there other equally realistic or better models with which to feed the upper level KMCM simulation? All that is required here is a commment on whether or not the same similarity features mentioned here are present in another modern tidal GCM. [If the tide is considered migrating only, then 6 hr data can yield phase and amplitude.] Does this other GCM have the parameters needed to feed the upper KMCM?

**2.** The figure arrangements are not conducive to comparison; some or all figure pairs should show meteor and KMCM together.

**3.** Semi-diurnal tidal phases should also be compared.

**4.** More detail is needed on special fitting process - e.g. what is the basic interval length, fitting method (least squares?), periods and fitting/subtraction order. Was a linear trend included in the fit?

**5.** Some significant meteor-KMCM differences have been glossed over, e.g. the KMCM tidal maximum in spring, which is not shown by the radar data. Figure 4 shows KMCM tide is almost linear at summer upper heights whereas meteor is probably circular; at least zonal and meridional components are roughly equal. Tidal phases are necessary to test circularity.

**6.** "special spectral ..." why are GW mentioned and not used. Unless the original data contains a pure sinusoid, when a spectral component is fitted and removed extra noise is created - that is, more is being subtracted than is actually present. An alternate method is to use an hourly difference filter, whose response peaks at 2 hr (but it also has a drawback in that there is a residual response at tidal frequencies.) There is additional noise in the original meteor wind fits created by angle-of-arrival and radial velocity errors (which, because of the radial nature of the measurements, cannot be broken down into zonal and meridional wind errors.)

Line 15-22: Explain how tidal variabilities "contaminate" the data. Is the argument that the KMCM does not have as much variability as meteor tidal amplitude data?

Line 13: Which references used which model(s)?

---

## Author Comment (AC1) · 27 Mar 2018

We thank the Referee for raising important issues. Below is our detailed response to the comments. We also highlighted changes in the text and attached the modified version as a supplement.

The first issue raised is the enhancement of semidiurnal tides is observed during the fall transition (September), while the model produces more seasonally symmetric climatology with the enhancement of semidiurnal tides during the fall transition and the lesser enhancement during the spring transition (April-May). Observationally, the tidal enhancement during the fall transition at high and middle latitudes is well reported

(e.g., Manson et at, 2009; Jacobi et al., 1999; Jacobi, 2012), as well as the fact that at lower latitudes the tidal climatology becomes more seasonally symmetric, with the enhancements both during the spring and the fall transitions (see Fig.1 attached below to the response showing semidiurnal tide climatology from the Canadian meteor radar, CMOR, at 43 deg N). To our knowledge, there is no definite theoretical explanation for the fall tidal enhancement being dominant at higher latitudes and, consequently, it is difficult to address this deficiency of the first-principle model. A possible explanation would be a height/slope of the mesospheric wind reversal boundary during May-August leading to stronger tidal amplification near the fall transition. Since the simulated wind reversal boundary is located somewhat lower and is less inclined (from May to August) comparing to the observations (see Fig. 5 in the article), one should expect more seasonally symmetric tidal climatology in the model. If the effect is related to the tidal amplification through interactions with mean flow and GWs in the MLT region, the new KMCM simulations with resolved GWs are likely to clarify the issue. This would be addressed in future studies. We added an extra discussion of this in the article.

The second issue raised is the overall enhancement of model tides above 85 km from June to September. We point out that the increase of tidal amplitudes above 85 km in summer is also seen in the observations in June-August, especially over Juliusruh radar (see Fig. 2 in the article, both zonal and meridional components), though to less extent than in the model. Again, the discrepancy is likely to be due to the lower height of mean flow reversal and to stronger summer zonal winds seen in the model. We commented on this in the text.

Regarding technical corrections: Page 2 line 19: The adaptive spectral filtering algorithm has been earlier described by the co-authors of the current paper (Stober et al., 2017), though its application to the extraction of tidal climatologies has not been previously published. We removed the term "novel" from the text/abstract, but this is a unique method for extracting tides, developed in our group. Page 2: We removed the footnote. Page 2, line 25: The IAP stands for the Leibniz-Institute of Atmospheric

Physics, we clarified this. Page 2 lines 31-32: The reference to Stober et al., 2017 is already in the text, but we also think it is useful to have one sentence briefly describing the method. Page 3 line 27: this sentence refers to the difference between modelled and observed dynamics, which is addressed earlier.

Please also note the supplement to this comment:
https://www.ann-geophys-discuss.net/angeo-2018-17/angeo-2018-17-AC1-supplement.pdf

[Figure]

[Figure]

**Fig. 1.** Amplitudes of semidiurnal tides extracted from the Canadian meteor data (43 deg N) using the same adaptive filtering technique.

**Supplement:**

[revised manuscript text omitted]

---

## Author Comment (AC2) · 4 Apr 2018

Below is our detailed response to the second referee's comments. The changes in the text are highlighted in yellow. Note that the changes are tracked with respect to the version modified after the first referee's comments. Numbers of sections below correspond to sections in the referee's report.

1. We have to point out that the paper is prepared for the AnnGeo Communicates section and has to be limited to 4 journal pages. The paper is thus bound to be focused on a limited number of issues. The main purpose is to show that the KMCM model provides reasonable description of thermal tides, consistent with radar observations, and thus this model can be used to force the ionospheric circulation model (TIEGCM) from below. This purpose is clearly stated in paragraph 10-15, page 4 and in the abstract. A direct comparison of KMCM with other atmospheric GCMs, while interesting, is way beyond the scope of this short paper. We added extra clarification of the study purpose in the text. We already included a brief review of earlier modelling results. Other modern GCMs produce similar climatologies of semidiurnal tides but, to our knowledge, all models have certain deficiencies, e.g., CESM/WACCM is known to produce substantially weaker tides (e.g., Smith, 2012). GSWM mentioned by the referee does not account for important processes such as nonlinear interactions with GWs and PWs, which is already noted in the paper. The main advantage of KMCM is in its simplified mechanistic character which makes it more suitable for the forcing of ionospheric GCM from below and for conducting numerical experiments. We have included further clarification of this in the text.

2. Fig. 5 already shows meteor radar data and KMCM together. Unfortunately the short paper format does not allow us to add extra figures.

3. We agree in principle that the tidal phases are important to analyse. However we have to leave this for a future study due to the length limitations.

4. This comment conflicts with the first referee's suggestion to remove all the details of fitting procedure and only refer to Stober et al., 2017 paper, so we have to compromise. The fitting method is least squares, further details are described in Stober et al., 2017. The length of sliding window is 3 days, we added in the text. The linear trend is fitted in this procedure and subtracted, we added the clarification. Fitted tidal periods are 24hr and 12hr, subtracted in this order.

5. We have now included extra discussion on the tidal amplification in spring seen in the KMCM simulations, as it was also requested by the first referee. The analysis of phases we believe should be left for another study, due to the length limitations of the article.

6. The main topic of the paper is a comparison of tides between meteor radar observations and the KMCM model. There are several other studies using meteor radars to investigate the GW seasonal properties (e.g., Hoffmann et al., 2010), so that we did not want to include and discuss these waves in the submitted manuscript. The used spectral filtering accounts for the full error propagation of the radial velocities plus iterative solution of the non-linear errors. In so far, we add no further noise to the derived quantities. The error due to angle of arrival is also accounted in our wind retrieval. The phase calibration of the meteor radar is checked using the astronomical position of meteor showers. We do not agree to the referee's comment that a radial velocity error "due to its radial nature" cannot be transferred to the zonal and meridional wind. In fact, this is mathematically included in our retrieval by making use of the covariance matrix.

Line 15-22: Variabilities do not contaminate the data but could make comparisons with models inconclusive. For instance, in the case of Davis et al., 2013, the natural short-term tidal variabilities, combined with radar measurement errors, are included into monthly variabilities (order of few m/s), meaning the modelled mean tidal amplitudes, both from CMAM and from WACCM models, generally fall in between the variability bars (see Fig. 10 in Davis et al., 2013).

Line 13, Page 2: Mitchell et al., 2002 used GSWM; Davis et al., 2013 used CMAM and WACCM. The text has been clarified.

Please also note the supplement to this comment:
https://www.ann-geophys-discuss.net/angeo-2018-17/angeo-2018-17-AC2-supplement.pdf

**Supplement:**

[revised manuscript text omitted]

---

## Editor Comment (EC1) · H. Lühr (Editor) · 6 Apr 2018

Dear Authors,

Both reviewers requested major or substantial revisions of your manuscript before decission on publication. You provided answers to all the comments. Now I ask you to resubmit a revised version of the manuscript to the Copernicus system. Please mark the relevant changes, if possible with different colours for the two reviewers.

This version is planned to go through another review cycle.

Best regards, Hermann Lühr

(Guest Editor)

---

## Referee Report (RR1)

Review(Reviewer # 2) of revised "Seasonal variability of atmospheric tides in the mesophere an lower thermosphere: meteor radar data and simulations" by Dimitry Pokhotelov et al. (ANGEO communicates, April 2018)

With regard to my first review - I was not informed that this paper format had a four page limit. My "light" comment still stands - but I agree can't be satisfied in four pages. The paper is now acceptable if the figure scales are equalized.

The response misunderstood what I said in the first review: I would re-word that comment as "an error in one (zonal or merid.) wind component *is* spread to the other because of the radial nature of the measurement". Yes, a standard least squares fit produces mathematical N and E error estimates but they are incorrect. Imagine that there are only variations in $V_E$, while $V_N$ is constant. A change in $V_E$ modifies the radial velocity, and thus modifies the $V_N$ component of the radial velocity. Thus, with radial measurement, the N and E wind compoents are coupled. Try a model. (I have). I think that explains why the on-line h90-format SKiYMET data do not show errors.

Figs. 1,2 (meteor radars) have one scale (0-60 m/s) while 3,4 (KMCM) have a different scale (0-50 m/s). For the stated purpose of comparison, the scales should be the same.

Fig. 5 is a good addition to the paper.

minor:

Pg 4, line 7:" are needed "

Pg 4, line 10 "to the model, where GWs are parameterised."

---

## Author Response (AR2)

Dear Editor,

Below we provide a detailed response to the Reviewer's comments. As the Reviewer suggested, we have modified the colour scales of Figs. 3 and 4.

Following your suggestion we added additional statements about model-data discrepancies, both in the Abstract and in the Summary. A revised version of the manuscript is attached at the end, with changes highlighted in green.

Best regards,

Dimitry Pokhotelov

**Response to the Anonymous Reviewer #2**

We followed the reviewer's suggestion and changed the colour scale of Fig. 3 and 4 to match the colour scale of Fig. 1 and 2 (0-60 m/s). We also made 2 minor corrections suggested by the reviewer.

Regarding the issue of error estimates, we have somewhat different prospective from the reviewer, as we summarised below. This is clearly beyond the scope of the current paper and could be discussed in a future work.

Indeed, there are two issues talking about errors. There are uncertainties related to the measurement itself, e.g., how good we can measure the radial velocity, the range or the angles of arrival and there are errors associated to the assumed model, which is used to obtain the winds. In the case of the meteor radar we have a large observation volume and apply an hour averaging assuming relatively constant wind velocity during this time.

The errors obtained from the retrieval are correct in a sense that they represent the statistical uncertainty of the radial velocity, range and angle of arrival uncertainties, but they do not provide an information whether the applied model was good, i.e. whether the wind can be assumed constant within one hour and within the observation volume.

In order to access at least a first order level of how good our assumptions are, the applied retrieval estimates the temporal and vertical shear and includes this as an additional error source in the balance of errors. In principle, we are able to derive and estimate the impact of each individual meteor to the final solution and as expected by the reviewer these shear terms in time and space, indeed have a significant error contribution to the final result.

The errors that we derived present at least a lower boundary of the uncertainties associated to the applied wind retrieval. As we already add shear terms in space and time, we even have a first order approximation of how good or bad our assumption of a constant wind field within each time and altitude bin is. These errors are propagated for each component and used as regularisation constrain in the retrieval.

In other words our solutions are not only determined by a least square for each time and altitude bin to reduce the impact that the reviewer mentioned in his comment. A more detailed discussion on what we actually do in our analysis is given in Stober et al., 2018 (AMT). However, these details hardly change the results of the climatology of the obtained tides. We tested different analysis schemes and temporal resolutions and the tidal climatology did not change a lot.

[revised manuscript text omitted]